# Association Between Maternal Diet and Frequency of Micronuclei in Mothers and Newborns: A Systematic Review

**DOI:** 10.3390/nu17152535

**Published:** 2025-08-01

**Authors:** Anny Cristine de Araújo, Priscila Kelly da Silva Bezerra do Nascimento, Marília Cristina Santos de Medeiros, Raul Hernandes Bortolin, Ricardo Ney Cobucci, Adriana Augusto de Rezende

**Affiliations:** 1Nutrition Postgraduate Program, Center for Health Sciences, Federal University of Rio Grande do Norte, Natal 59078-900, RN, Brazil; annycristinearaujo@gmail.com (A.C.d.A.); priscilaksb@yahoo.com.br (P.K.d.S.B.d.N.); 2Health Sciences Postgraduate Program, Center for Health Sciences, Federal University of Rio Grande do Norte, Natal 59078-900, RN, Brazil; marilia066@hotmail.com (M.C.S.d.M.); rhbortolin@gmail.com (R.H.B.); 3Sciences Applied to Women’s Health Postgraduate Program, Center for Health Sciences, Maternidade Escola Januário Cicco (MECJ/EBSERH), Federal University of Rio Grande do Norte, Natal 59078-900, RN, Brazil; ricardo.cobucci.737@ufrn.edu.br; 4Department of Clinical and Toxicological Analyses, Federal University of Rio Grande do Norte, Natal 59078-900, RN, Brazil

**Keywords:** mother–child relationships, micronucleus test, maternal diet, systematic review

## Abstract

**Background/Objectives:** The effect of diet on maternal and infant genetic levels has been reported in the literature. Diet-associated DNA damage, such as the presence of micronuclei (MN), may be related to an increased risk of developing chronic diseases such as cancer. There is particular concern regarding this damage during pregnancy, as it may affect the newborn (NB). Thus, this review aims to summarize the primary evidence of the impact of diet on the frequency of MN in the mother–infant population. **Methods**: Five databases (PubMed, Embase, Web of Science, Scopus, and ScienceDirect) were used to search for observational studies. Google Scholar and manual searching were required to perform the “gray literature” search. **Results**: The search strategy retrieved 1418 records. Of these, 13 were read in full and 5 were included in the review. Most studies were of the cohort type (*n* = 4) and were carried out in the European region. A total of 875 pregnant women and 238 newborns were evaluated. Despite insufficient evidence to confirm that diet changes the frequency of MN, the included studies found possible effects from the consumption of fried red meat and processed meats and the adequate consumption of vegetables and polyunsaturated fats. **Conclusions**: Future research is needed in order to understand the effects of diet on genetic stability and to obtain evidence to help plan public policies on food and nutrition or reinforce protective dietary patterns for this and future generations.

## 1. Introduction

Micronuclei (MN) are chromosomal fragments and/or whole chromosomes that fail to reincorporate into the nucleus during cell division [1]. MN are widely recognized as one of the most established biomarkers of chromosomal damage and genetic instability, and their occurrence has been extensively studied [2]. The cytokinesis-blocked micronucleus assay (CBMN), developed in 1960 [3], is a well-established method frequently employed in human biomonitoring studies to assess genotoxicity from environmental or endogenous factors. Importantly, this assay correlates with future health risks in adults, including cancer [4,5].

Over time, this assay has evolved to detect additional cellular alterations beyond MN, such as chromosome breakage, chromosome loss, non-disjunction, necrosis, apoptosis, and cytostasis [6]. This test is an effective tool for investigating cellular and nuclear dysfunctions arising from aging, micronutrient imbalances, and exposure to genotoxic compounds. It holds promise for applications in emerging fields, such as nutrigenomics and toxicogenomics, further supporting the hypothesis that nutritional factors can influence sensitivity to environmental genotoxins, including those from dietary sources [1,7].

In this context, the Human Micronucleus Biomonitoring Project (HUMN.org) was launched to build a global database and establish MN frequencies in human populations worldwide [8]. After 26 years of activity, the HUMN project provided the initial evidence that MN frequency may be linked to reduced reproductive capacity [9]. In pregnant women, a high frequency of MN in lymphocytes has been associated with complications such as spontaneous abortion, intrauterine growth restriction, and preeclampsia. Moreover, this project emphasized the need to explore the relationship between diet and MN frequency [6,9].

Maternal diet is a modifiable factor that is increasingly being recognized as crucial for both maternal and fetal health [10]. Since 1997, studies investigating the relationship between food consumption and MN frequency have emerged, with a growing interest in understanding the impact of fruits, meat, and vitamins on this biomarker [11,12,13]. However, the association between maternal diet, MN frequency, and its effects on pregnant women and newborns remains unclear [6,14]. Addressing this gap is vital, particularly because the consumption of dietary carcinogens can influence pregnancy outcomes [15]. Nutritional factors have been widely discussed in the scientific community, as it is believed that an adequate diet can supply essential vitamins and minerals, thus supporting proper DNA synthesis and repair mechanisms [16,17].

In recent years, several studies have explored and demonstrated the associations between nutrient intake and genetic damage, whereas others have reported inconclusive results [18,19,20]. Therefore, this systematic review aimed to compile and critically analyze the available evidence regarding the effects of maternal diet on MN frequency in mothers and their newborns. This analysis can guide future research and enhance our understanding of nutrition and public health issues.

## 2. Materials and Methods

The review protocol was registered on the International Prospective Register of Systematic Reviews (PROSPERO) platform under the registration number CRD42023422903 and has been published previously [21]. This review followed the guidelines for the Preferred Reporting Items for Systematic Reviews and Meta-Analyses [22].

### 2.1. Eligibility Criteria

The eligibility criteria were based on the PECOS strategy (population, exposure, comparison, outcome, and study design). The population (P) included pregnant women. Exposure (E) referred to an inadequate diet regarding the consumption of food groups, macronutrients, and micronutrients, according to the recommendations of the World Cancer Research Fund and the Dietary Reference Intakes, respectively. Comparison (C) comprised an adequate diet, according to the same recommendations. Moreover, outcome (O) was the frequency of micronuclei quantified by the CBMN and/or buccal micronucleus cytome assay (BMCyt) in oral mucosa cells, peripheral blood of pregnant women, and/or umbilical cord blood of newborns. Observational studies with cross-sectional, case-control, or cohort designs that investigated food consumption via food frequency questionnaires were included. Observational studies that evaluated the frequency of MN in pregnant women with comorbidities such as gestational or preexisting diabetes, chronic hypertension, or pregnancy-induced hypertension and in children older than 28 days were excluded.

### 2.2. Search Strategy

The search strategy was developed by considering terms registered in the Medical Subject Headings platform, the most frequent terms in scientific articles, and Boolean operators. It was adapted for each database according to the specific guidelines of each platform (Appendix A). The following databases were used: PubMed, ScienceDirect, Web of Science, Embase, and Scopus. When available, filters were applied to restrict results to human studies. A gray literature search was performed using Google Scholar, and the first 200 records were considered [23]. Additionally, a manual search of the references of the articles included in this review was conducted. The search covered records available until January 2025. Language restrictions were not imposed.

### 2.3. Selection of Studies

Using Rayyan software (version 10.10.98), three independent reviewers (ACA, MCSM, and PKSBN) evaluated all abstracts retrieved from the databases. Disagreements were resolved by a fourth reviewer (A.A.R.). The following data were extracted from the eligible records using a standardized form: author, year of publication, study location (country), study design, sample size, frequency of micronuclei in pregnant women and newborns, method of micronuclei quantification, statistical analysis, and primary results.

### 2.4. Data Synthesis and Analyses

The results of the studies included in this systematic review are synthesized and presented as a narrative summary. The results included the type of test used to measure MN, maternal diet consumed, and outcomes observed in the target population. The included studies exhibited considerable variability in the manner in which they reported data on MN frequency. To determine the total number of participants, the mean was calculated from studies that used data from the same cohorts. The total sum was then obtained. This approach reduced the risk of data duplication. The heterogeneity of the studies makes it inappropriate to conduct meta-analyses.

### 2.5. Assessment of Risk of Bias

Three independent reviewers (A.C.A., M.C.S.M., and P.K.S.B.N.) used the five-item Newcastle–Ottawa Scale [24] to assess the risk of bias in cohort studies. An adapted version with seven questions was used to evaluate the cross-sectional studies. Quality ratings were based on the final score; studies rated as high quality received six or more stars, those rated as moderate quality received four or five stars, and those rated as low quality received fewer than four stars.

## 3. Results

### 3.1. Selection of Studies

In this systematic review examining the frequency of MN in pregnant women and their newborns, 1418 records were retrieved through database searches and other methods (Figure 1). After excluding 205 duplicates, 1213 records were retained for title and abstract screening. Thirty records were reviewed in full and assessed against the eligibility criteria, resulting in the inclusion of five studies in this review [25,26,27,28,29] (Appendix A). None of the articles permitted a meta-analysis because of variations in the presentation of the assessed dietary nutrients or the high heterogeneity of the results obtained.

### 3.2. Characteristics of Observational Studies

Characteristics of the five studies are presented in Table 1. Of the five records, four were cohort studies, and one was cross-sectional [29]. These studies were conducted in Greece (*n* = 3) [26,27,28] and Denmark (*n* = 1) [29]. One study included cohorts from the United Kingdom, Spain, Norway, Greece, and Denmark [25]. No such studies have been conducted in Brazil or other South American countries. The included studies reflected an average publication age of 7.4 (±2.7) years, with only one published in the last five years.

### 3.3. Frequency of Baseline MN in Pregnant Women and Newborns

In total, 875 pregnant women and 238 newborns were included in this study. Notably, the study by Loock et al. [25] was the only study included in this review that did not assess newborns.

Overall, the MN frequencies were consistently reported as means or medians across all studies. The highest MN frequencies were reported by Pedersen et al. [29] in Denmark, where the authors identified a median maternal frequency of 6.97 cells with MN‰ and a median frequency of 3.16 cells with MN‰ in newborns. The lowest maternal MN frequency was observed in a study by O’Callaghan-Gordo et al. [28], in which women exhibited a frequency of 2.39 MN‰. Meanwhile, another study by O’Callaghan-Gordo et al. [27] reported the lowest MN frequency in newborns, at 1.44 MN‰.

#### 3.3.1. Method for Measuring MN Frequency

The included studies performed the CBMN assay following the methods of Decordier et al. [30] using maternal peripheral blood and umbilical cord blood from newborns. MN scoring was performed using a semi-automated image analysis system [30]. No study conducted BMCyt testing.

#### 3.3.2. Maternal Diet Assessment

Three studies administered the Food Frequency Questionnaire during the 14th to 18th weeks of pregnancy. Food consumption questionnaires were distributed for 250–431 food items. Generally, the included studies [25,26,27,28,29] employed images to differentiate portion sizes of each food item and used standard recipes for mixed and complex dishes. Only one study conducted in Denmark [29] assessed cooking methods for meat products and employed software to quantify food intake.

The frequency of MN was assessed in relation to fruit and vegetable intake (*n* = 3) [28] and micronutrient consumption (*n* = 2), specifically vitamin D [27] and vitamin C [26]. Only Pedersen et al. [29] evaluated the frequency of MN in relation to the dietary intake of macronutrients, energy, and fiber. Additionally, we investigated the relationship between MN frequency and the consumption of chemical compounds formed during the cooking process of ingested foods. Similar to Pedersen et al. [29], Loock et al. [25] evaluated macronutrients but focused exclusively on fat consumption. The data retrieved from our search were insufficient for quantitative analysis because the way in which they were reported and/or grouped hindered the ability to conduct a meta-analysis.

#### 3.3.3. Statistical Analysis

The five analyzed studies employed different statistical approaches to investigate the relationship between maternal diet and MN frequency. Parametric and non-parametric analyses were used, including the analysis of variance, *t*-test, and Kruskal–Wallis test, as well as regression models such as negative binomial regression with incidence rate ratio (IRR) estimates and 95% confidence intervals, along with relative risk (RR) and their respective confidence intervals. These methodologies allowed for the estimation of the relationship between dietary factors and MN frequency, adjusted for confounding variables. The effect size was not reported in any of these studies. The main findings are summarized in Table 2.

### 3.4. Effect of Diet on the Frequency of Micronuclei

#### 3.4.1. Pregnant Women

Table 2 shows the intake of the dietary components assessed in the studies and their effects on the frequency of MN in pregnant women and newborns. Studies conducted by Pedersen et al. [29] and O’Callaghan-Gordo et al. [26,28] evaluated the frequency of MN based on food consumption rate (g/d) in each group. None of the included studies identified a significant effect of an adequate vegetable-based diet on MN frequency in pregnant women (Table 2). O’Callaghan-Gordo et al. [28] observed an increase in the MN frequency in mononuclear cells when dietary intake was compared across tertiles. Women with a vegetable intake of 152 to 251 g/d had a higher relative risk (IRR = 1.75 [1.09, 2.82]) for increased MN frequency compared to those with an intake of 251 to 1039 g/d (IRR = 1.28 [0.80, 2.05]) [28]. In another study, O’Callaghan-Gordo et al. [26] similarly subdivided fruit and vegetable intake into tertiles but reported no difference in MN frequency between groups. O’Callaghan-Gordo et al. [28] also analyzed data from a Greek cohort, revealing that pregnant women generally had an adequate intake of red meat but an inadequate intake of processed meat. Only red meat consumption was associated with an increased MN frequency in mothers (Table 2), with elevations noted in both the 2nd tertile (19 to 51 g/d; IRR = 1.34 [1.00, 1.80]) and the 3rd tertile (51 to 221 g/d; IRR = 1.33 [0.96, 1.85]) [28]. Similarly, a cross-sectional study by Pedersen et al. [29] in Denmark demonstrated a predominance of high-protein diets, particularly fried meat with darkened surfaces; however, this was not associated with MN frequency in mothers. Nonetheless, the study reported significant associations between dietary preferences and other genotoxicity assays [29].

Although Pedersen et al. [29] evaluated macronutrient intake, no significant differences in MN frequency were observed between mothers and newborns. Conversely, Loock et al. [25], in their cohort study, assessed the lipid intake of 625 women across different European countries and found a dietary pattern rich in omega-3 and omega-6 fatty acids. Their analysis revealed a protective effect, with a 1 g/day increase in omega-6 intake reducing MN frequency by 6% (IRR = 0.99; *p* = 0.047), whereas a 1 g/day increase in omega-3 intake reduced MN frequency by 3% (IRR = 0.97; *p* = 0.047). Other dietary components did not significantly affect MN frequency.

Two additional studies using cohort data from Crete, Greece, assessed the effects of micronutrients on women and their newborns [26,27]. In the first study [26], diets with adequate or inadequate vitamin C intake showed no significant differences in maternal MN frequency (Table 2). However, O’Callaghan-Gordo et al. [26] found that insufficient vitamin C intake coupled with high air pollution levels significantly increased MN frequency (RR = 5.57 [1.96, 15.81]; *p* = 0.004). In the second study, O’Callaghan-Gordo et al. [27], found that 73% of women had a diet deficient in vitamin D, with a mean intake of 1.9 (1.4) µg/day. However, no association was found between vitamin D intake and MN frequency. In both studies [26,27], the small sample sizes limited the ability to accurately determine the potential effects of micronutrient intake and draw definitive conclusions regarding its impact on genetic damage.

#### 3.4.2. Newborns

The included studies did not reveal statistically significant differences in MN frequency between newborns with adequate and those with inadequate dietary intake patterns. However, O’Callaghan-Gordo et al. [28] observed lower MN values in the children of women with higher vegetable intake when comparing consumption tertiles (2nd tertile: IRR = 0.56 [1.34, 0.91]; 3rd tertile: IRR = 0.63 [0.39, 1.00]). A cross-sectional study conducted by Pedersen et al. [29] indicated a probable association between the maternal preference for fried meat with dark surfaces and increased MN frequency in newborns. However, this association was not statistically significant (*p* = 0.7). The small sample size and recall bias in estimating food intake may have been limiting factors in this study. In contrast, a cohort study by O’Callaghan-Gordo et al. [28] found that maternal red meat consumption was associated with lower MN frequencies in newborns (Table 2). Additionally, newborns of women with an inadequate consumption of processed meat exhibited a higher frequency of MN, as demonstrated when comparing consumption tertiles (2nd tertile: IRR = 1.70 [1.12, 2.58]; 3rd tertile: IRR = 1.95 [1.23, 3.10]) [28].

Among the five included studies, two examined the effect of maternal micronutrient intake on MN frequency in newborns, both conducted by O’Callaghan-Gordo et al. [26,27]. No significant relationships were identified between inadequate dietary vitamin C intake in mothers and MN frequency in newborns. Any observed associations were deemed to be due to chance and lack of biological plausibility [26]. Conversely, an association has been found between inadequate dietary vitamin D intake by mothers and increased MN frequency in newborns (Table 2) [27].

### 3.5. Methodological Quality of Studies

The average quality score of the studies was six stars. The most challenging items to assess were the representativeness of the exposed cohort and the adequacy of cohort monitoring. These studies did not justify the calculation of sample sizes related to these items. Additionally, 60% (*n* = 5) of the studies experienced a sample loss of >20% during the execution of the study. Among all the evaluated studies, 60% (*n* = 3) were classified as displaying high methodological quality (≥6 stars), whereas 40% (*n* = 2) were deemed to show moderate quality (≥4 and ≤5 stars). The methodological quality assessment is detailed in Appendix A.

## 4. Discussion

This systematic review underscores the scarcity of studies assessing the association between dietary patterns and MN frequency in the maternal–infant population. Although evidence suggests a potential link between diet and genomic instability, several methodological limitations limit the strength of these findings.

The methodological quality of the included studies was heterogeneous, with a predominance of high-quality research. However, critical limitations were identified, including a lack of justification for the sample size and significant sample attrition (>20% in 60% of the studies), raising concerns about selection bias and statistical power.

Additionally, dietary assessment methods varied considerably among the included studies and were predominantly based on self-reported intake. The findings presented should be considered exploratory and hypothesis-generating. Most studies relied on self-reported data, such as food frequency questionnaires, which exhibit inherent limitations related to memory, perception, and reporting accuracy. These tools are susceptible to systematic errors, including under- or overestimation of consumption, potentially introducing misclassification bias and undermining the reliability of observed associations between maternal diet and micronuclei frequency.

Although this review describes associations as reported by the authors of the included studies, it is important to note that in several cases, the confidence intervals for the effect estimates included the null value, indicating statistical non-significance. This highlights a limitation in the strength of the evidence and the need for cautious interpretation, as these findings may reflect insufficient power or imprecise estimates rather than a true absence or presence of effect.

To improve the accuracy of dietary evaluation, future research should consider using repeated 24 h dietary recalls at multiple time points, which may enhance the reliability of dietary pattern analysis. Additionally, integrating objective nutritional assessment methods, such as serum biomarkers of vitamins and other micronutrients, could help validate self-reported intake and clarify potential causal relationships between maternal nutrition and genomic instability in mothers and newborns.

The absence of standardized nutritional assessment approaches further limits comparability across studies and weakens the ability to draw definitive conclusions. Two systematic reviews published in 2021 [31,32] also highlighted these methodological concerns and reinforced the need for validated dietary assessment tools and objective biomarkers to enhance data reliability and facilitate cross-study comparisons.

Beyond methodological constraints, the biological plausibility of diet-induced MN formation requires further investigation. Some studies have reported an increased MN frequency associated with the consumption of processed and red meat, a finding consistent with that in non-pregnant populations [33]. This effect is likely mediated by exposure to mutagenic compounds such as polycyclic aromatic hydrocarbons and heterocyclic amines, which are formed during high-temperature cooking [34,35]. Conversely, evidence from one study suggested a protective effect of diets rich in fruits, vegetables, and polyunsaturated fatty acids, potentially due to their antioxidant and anti-inflammatory properties [36]. However, given the limited number of available studies, these findings remain preliminary, and further research is required to establish their causality and clinical relevance.

Another key consideration is the potentially confounding effects of environmental exposure. Few of the included studies adequately accounted for external factors such as air pollution, pesticide residues, or heavy metals, all of which have been implicated in DNA damage and MN formation. As many of the reviewed studies were conducted in urban settings, the impact of these exposures cannot be overlooked. A more comprehensive understanding of the interplay between environmental and dietary factors regarding genomic stability can be achieved through multivariate approaches that integrate these variables into future research.

Although diet is known to affect various biological processes, the evidence presented in this review remains insufficient to determine whether any specific dietary pattern significantly alters MN frequency in the maternal–infant population. Furthermore, our findings highlight the need for well-designed prospective studies with rigorous methodological frameworks, larger sample sizes, and standardized dietary assessments. Moreover, incorporating complementary genomic stability biomarkers such as comet assays and epigenetic analyses could enhance the robustness of future investigations and provide deeper insights into the mechanisms linking diet and genomic instability during pregnancy.

Recent literature increasingly highlights the role of epigenetic mechanisms in mediating the effects of maternal nutrition on fetal development, especially during critical windows such as pregnancy and early infancy. Evidence suggests that maternal intake of nutrients like polyunsaturated fatty acids, vitamins, and bioactive compounds from breast milk may modulate DNA methylation and gene expression pathways related to immune regulation and cellular stability, including T cell differentiation and metabolic reprogramming [37,38]. These epigenetic modifications, which are heritable and reversible, may influence long-term health outcomes, including the risk of allergic and chronic diseases. Although primarily focused on immune-related outcomes, these studies support the plausibility that maternal diet could similarly impact genomic stability via epigenetically mediated pathways.

Integrating such epigenetic insights with cytogenetic markers like MN may be instrumental in uncovering the biological underpinnings of nutritional programming and its effects on chromosomal integrity. Therefore, future studies should consider a broader panel of molecular and functional endpoints to elucidate these complex interactions and identify potential windows of susceptibility.

This review displays several strengths. It combines well-characterized studies of MN frequency using strong statistical methods. Notably, it is the first to examine the link between dietary intake and MN frequency in mothers and infants. This study offers valuable insight into previously overlooked areas. However, this study exhibits significant limitations. The small number of available studies and the small sample sizes of most cohorts limit the generalizability of the findings. Another limitation is the use of different dietary assessment methods. This, along with the lack of standardized protocols, makes a meta-analysis impossible. These factors highlight the need for future studies with larger sample sizes, better methods, and standardized dietary assessment tools. This will help us to understand how diet affects genomic stability during pregnancy.

## 5. Conclusions

This systematic review highlights the limited and heterogeneous evidence regarding the relationship between maternal diet and MN frequency in pregnant women and newborns.

Although some studies suggest a possible link between a high intake of processed meats and increased MN formation, as well as a protective effect of polyunsaturated fats, methodological inconsistencies hinder definitive conclusions. Furthermore, environmental factors known to influence genomic stability were not adequately addressed in most studies.

To advance understanding in this field, future research should adopt more rigorous designs, including adequately powered samples based on statistical calculations and the use of validated dietary assessment instruments, such as repeated 24 h dietary recalls and detailed food records. Combining these tools with objective biomarkers, such serum levels of critical micronutrients during pregnancy, can enhance the accuracy of nutritional exposure assessment. Evaluating genomic integrity through complementary assays, such as comet assays, DNA methylation profiling, and expression analysis of genes involved in DNA repair and cell cycle regulation, may provide deeper insight into the biological mechanisms underlying MN formation.

Additionally, randomized controlled trials, which represent a high level of evidence in the scientific hierarchy, could minimize potential biases and help establish causal relationships. Future investigations should also incorporate multivariate models to account for confounding factors such as environmental exposures and maternal lifestyle, thereby isolating the specific effects of maternal diet on chromosomal stability during pregnancy.

## Figures and Tables

**Figure 1 nutrients-17-02535-f001:**
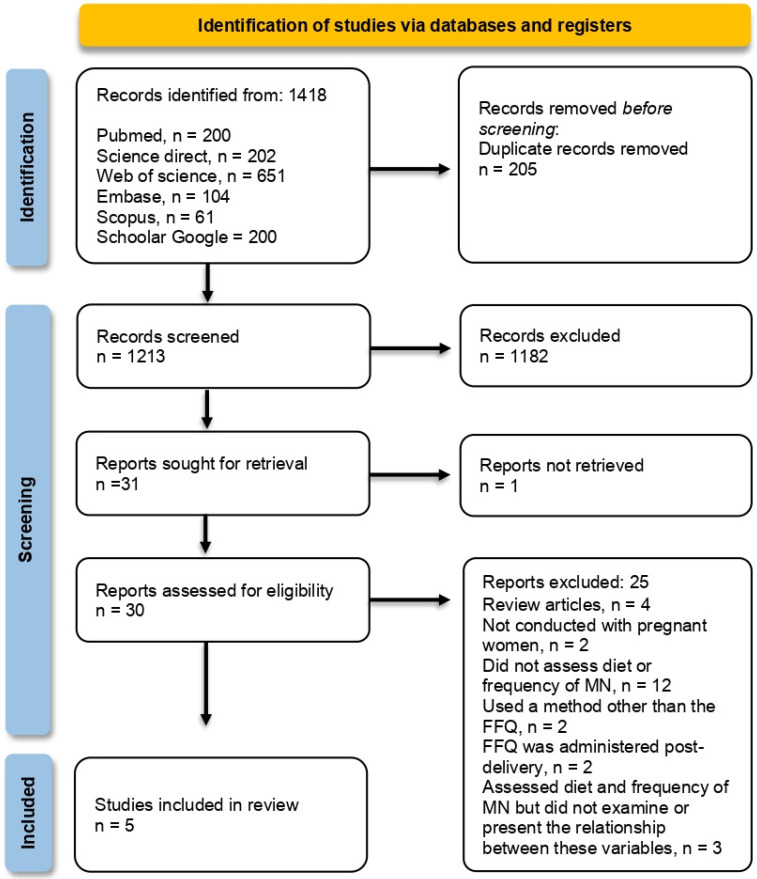
Flow chart of the study screening process based on the PRISMA protocol.

**Table 1 nutrients-17-02535-t001:** Characterization of the studies included in this review (*n* = 5).

Author (year)	Study Design	Country	Total Number		MN Frequency (‰)		MN Quantification Method	Analyzed Dietary Component	Statistical Method	Main Results
			Pregnant Women	Newborns	Pregnant Women	Newborns				
Loock et al. (2014) [25]	Cohort	United Kingdom, Spain, Norway, Greece, Denmark	625	-	2.53 (2.06) ^b^	-	CBMN	Macronutrients: fat (omega-3 and -6).	ANOVA, *t* test, Kruskal–Wallis, negative binomial regression, and estimated incidence rate (IRR; *p* value).	In pregnant women, consuming a diet with adequate omega-3 reduces (↓) the frequency of MN by up to 3% (IRR = 0.97; *p* = 0.047), and the consumption of 6 reduced the frequency of maternal MN by 6% (IRR = 0.94; 0.047).
O’Callanghan-Gordo et al. (2015) [26]	Cohort	Greece	181	183	2.63 (2.53) ^a^	1.48 (1.83) ^a^	CBMN	Food groups: fruits and vegetables; micronutrients: vitamin C.	Negative binomial regression, relative risk and 95% confidence interval [RR(95% CI)].	Reduced (↓) vitamin C intake shows increased (↑) frequency of MN when associated with other environmental factors [RR = 9.35 (2.77–31.61)].
O’Callanghan-Gordo et al. (2017) [27]	Cohort	Greece	173	171	2.41 (2.48) ^a^	1.44 (1.58) ^a^	CBMN	Micronutrients: Vitamin D.	Negative binomial regression, estimated incidence rates (IRR), and 95% confidence intervals.	Reduced (↓) maternal vitamin D intake increased (↑) the frequency of MN in the umbilical cord blood of newborns [mean tertile IRR = 1.51 (1.06–2.14)].
O’Callanghan-Gordo et al. (2018) [28]	Cohort	Greece	188	200	2.39 (2.42) ^a^	1.52 (1.65) ^a^	CBMN	Food groups: fruits, vegetables, red meats, processed meats.	Negative binomial regression, estimated incidence rates (IRR), and 95% confidence intervals.	Increased (↑) of red meat consumption was associated with ↑ frequency of MN in newborns [2nd tertile IRR = 1.34 (1.00, 1.80); 3rd tertile IRR = 1.33 (0.96–1.85)].
Pedersen et al. (2012) [29]	Cross-sectional	Denmark	69	54	6.97 (2.89–13.73) ^a^	3.16 (0.00–7.12) ^a^	CBMN	Food groups: cereals, fruits, vegetables, legumes, red meat, poultry, fish, milk, eggs, potatoes; macronutrients: energy, fiber, carbohydrates, protein, and fat.	Wilcoxon and negative binomial regression, and 95% confidence interval (β; 95% CI).	No association was observed between newborns of women who consumed fried meat increased (↑) and the frequency of MN, but with other types of DNA damage (β = 0.46; 0.08, 0.84).

Note: - values not identified; ^a^ median (95% confidence interval); ^b^ mean (standard deviation); MN, micronuclei; CBMN, cytokinesis-blocked micronucleus assay in lymphocytes; IRR, relative risk incidence; RR, relative risk.

**Table 2 nutrients-17-02535-t002:** Main results extracted from the included studies.

Author (Year)	Pregnant Women							Newborns					
		Adequate Diet		Inadequate Diet				Adequate Diet		Inadequate Diet			
	Assessed Dietary Components	*n*	MN	*n*	MN	Relative Risk(95% CI)	Effect on the Frequency of MN	*n*	MN	*n*	MN	Relative Risk(95% CI)	Effect on the Frequency of MN
O’Callanghan-Gordo et al. (2015) ^a^ [26]	Fruits and/or vegetables	41	2.9	40	3.0	-	No effect	47	1.5	41	1.1	-	No effect
O’Callanghan-Gordo et al. (2018) [28]		-	-	-	-	-	-	-	-	-	-	-	-
Pedersen et al. 2012 ^b^ [29]	Red meat	38	-	52	-	−0.2 (−0.4 to 0.0)	No effect	38	4	54	3	0.3 (−0.0 to 0.7) ^b^	No effect
O’Callanghan-Gordo et al. (2018) ^c^ [28]		41	1.4 (1.02, 1.92)	40	**1.47** **(1.01, 2.15) ***		**Inadequate diet led to an increase.**	46	**0.85 (0.6, 1.2)**	46	0.91 (0.64, 1.3)		**Adequate diet led to a reduction.**
O’Callanghan-Gordo et al. (2018) ^c^ [28]	Processed meat	34	0.89 (0.64, 1.22)	39	0.99 (0.72, 1.37)		No effect	43	1.46 (0.94, 2.26) ^d^	46	**1.69** **(1.06, 2.68)** ^d^ *****		**Inadequate diet led to an increase.**
Loock et al. (2014) [25]	Fats (omega-3)	-	-	-	-	**0.97 ***	**Adequate diet led to a reduction.**	NA	NA	NA	NA		
	Fats (omega-6)	-	-	-	-	**0.94 ***	**Adequate diet led to a reduction.**	NA	NA	NA	NA		
Pedersen et al. (2012) ^b^ [29]	Total fat	-	-	-	-	−0.1 (−0.4 to 0.2)	No effect	-	-	-	-	0.0 (−0.5 to 0.5)	No effect
	Carbohydrates	-	-	-	-	−0.1 (−0.2 to 0.1)	No effect	-	-	-	-	0.1 (−0.3 to 0.6)	No effect
	Protein	-	-	-	-	0.1 (−0.1 to 0.4)	No effect	-	-	-	-	0.1 (−0.4 to 0.5)	No effect
O’Callanghan-Gordo et al. (2015) ^a^ [26]	Vitamin C	69	2.4	20	2.4	-	No effect	22	1.1	62	1.4	-	No effect
O’Callanghan-Gordo et al. (2017) [27]	Vitamin D	42	1	50	0.96 ^c^	0.94 (0.67, 1.32) ^d^	No effect	43	1	45	1.14 ^c^	**1.51** **(1.06, 2.14)** ^d^*****	**Inadequate diet led to an increase.**

Note: * statistically significant *p*-values < 0.05 for the tests performed; ^a^ MN frequency values presented as median; ^b^ regression estimates presented as β value and 95% CI; ^c^ MN frequency values presented as incidence rate ratio and raw value, considering paired analysis; ^d^ MN frequency values presented as incidence rate ratio and adjusted value, considering paired analysis; - values not identified in the text or supplemental files. MN, micronuclei; Results in bold indicate statistical significance.

## Data Availability

The data that support the findings of this review are derived from publicly available sources. All included studies and datasets were obtained from published research articles, which are accessible via database searches. No new primary data were generated in the course of this research.

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
