# Peer review of "Association Between Maternal Diet and Frequency of Micronuclei in Mothers and Newborns: A Systematic Review"

_nutrients, 2025, doi:10.3390/nu17152535_

Round 1
Reviewer 1 Report
Comments and Suggestions for Authors
With interest, I read the manuscript entitled “Association between maternal diet and frequency of micronuclei: A systematic review” (submission ID: nutrients-3781185), written by Araújo and colleagues.
Although in the beginning, I had mixed fillings about this manuscript, I find it potentially interesting. My initial problems with this work were related to the fact that only five papers were included in the final analysis. On the other hand, the Authors are not guilty of the fact that only this number of papers were able to make it.
Thus, I have only several minor/facultative comments:
- Line 22, in the Abstract. The Authors write that They started with 4558 records, in all other sections They mention 1418 as a starting point. This difference should be clarified.
- The Authors focus on genetic influences on the pregnant women and as a result on the offspring. However, they should take into account the potential epigenetic influences into account (PMID: 33668787, 33193294) in the discussion.
- The graphical abstract would be an added value.
- In the discussion, the Authors should refer to animal studies, if any.
Author Response
We would like to thank the reviewer for the thoughtful and constructive feedback. The suggestions provided were extremely helpful in improving the clarity, consistency, and overall quality of our manuscript. We sincerely appreciate the time and effort dedicated to the review process.
Comments 1: Line 22, in the Abstract. The Authors write that They started with 4558 records, in all other sections They mention 1418 as a starting point. This difference should be clarified;
Response 1: We thank the reviewer for pointing out this inconsistency. Indeed, this was a typing error in the Abstract. The correct number of records initially identified was 1418, as consistently presented in the other sections of the manuscript. This has been corrected in the revised version. Line 22 of the Abstract now correctly states: “We initially identified 1418 records…”
Comments 2: The Authors focus on genetic influences on the pregnant women and as a result on the offspring. However, they should take into account the potential epigenetic influences into account (PMID: 33668787, 33193294) in the discussion.
Response 2: We appreciate this valuable suggestion. The recommended references provided important insights and helped enrich our discussion. We have now incorporated them in the section addressing maternal-fetal influences, highlighting the role of epigenetic mechanisms. This addition can be found in lines 354–364 of the revised manuscript. A new paragraph has been added to the Discussion section (lines 354–364) incorporating the epigenetic perspective and referencing the suggested articles.
Comments 3: The graphical abstract would be an added value.
Response 3: We thank the reviewer for the suggestion. A graphical abstract has now been created and submitted with this revised version of the manuscript, as recommended.
Comments 4: In the discussion, the Authors should refer to animal studies, if any.
Response 4: We appreciate the reviewer’s suggestion. However, as our review focuses specifically on human studies regarding the effects of maternal micronutrient status, we chose not to include findings from animal models. During our literature search, we did not identify relevant animal studies addressing the specific micronutrients discussed in our review.
Reviewer 2 Report
Comments and Suggestions for Authors
This systematic review addresses an important and timely topic: the potential association between maternal diet and genomic instability as measured by micronucleus (MN) frequency in pregnant women and their newborns. The manuscript is generally well-structured with comprehensive literature searching and appropriate data extraction. However, the strength of the evidence is limited due to the small number of included studies, methodological variability, and lack of meta-analysis.
While the findings are of interest and may contribute to future research directions in maternal nutrition and genetic health, several critical issues—particularly related to dietary exposure assessment, biological plausibility, statistical interpretation, and control of confounding—need to be addressed to strengthen the manuscript. I recommend major revision before the manuscript can be considered for publication.
Major concerns
- Only 5 observational studies were included, with no meta-analysis possible due to heterogeneity. The authors should more clearly state that the evidence is preliminary and mostly hypothesis-generating.
- The review mentions the use of FFQs and dietary recall, but it underplays the limitations of self-reported data. A critical discussion of nutrient intake quantification error and possible misclassification bias is needed.
- The review uses terms like “associated with” or “no effect” across studies, but in some cases, confidence intervals include 1.0 or cross zero, meaning non-significance.
Minor concerns
- Title clarity: “Association between maternal diet and frequency of micronuclei” → consider adding “in mothers and newborns” to make the scope clearer.
- Grammar: Some sections use passive voice excessively. Consider active voice for clarity (e.g., “was observed” → “researchers observed”).
- Data Table 2: Please highlight significant findings visually (e.g., bold or asterisk).
- Conclusion section: The call for further research is clear, but recommendations for future study design (e.g., sample size calculation, use of biomarkers) should be more specific.
Author Response
Comments 1: Only 5 observational studies were included, with no meta-analysis possible due to heterogeneity. The authors should more clearly state that the evidence is preliminary and mostly hypothesis-generating.
Response 1: We thank the reviewer for this important observation. We fully agree that, given the small number of included studies and the methodological heterogeneity, the evidence presented should be considered preliminary and hypothesis-generating. We have revised the manuscript accordingly to reflect this more clearly. This modification can be found in lines 301–308. A statement emphasizing the preliminary nature of the findings and their value in guiding future research has been added in the Discussion section (lines 301–308).
Comments 2: The review mentions the use of FFQs and dietary recall, but it underplays the limitations of self-reported data. A critical discussion of nutrient intake quantification error and possible misclassification bias is needed.
Response 2: We thank the reviewer for this valuable comment. We agree that self-reported dietary assessment methods, such as FFQs and 24-hour recalls, have important limitations that can affect the accuracy of nutrient intake estimates. In response, we have added a more critical discussion regarding potential quantification errors and misclassification bias associated with these tools. The revised text can be found in lines 315–324.
Lines 315–324 now include a discussion on the limitations of self-reported dietary data, including possible under- or overestimation of intake and the implications for study validity.
Comments 3: The review uses terms like “associated with” or “no effect” across studies, but in some cases, confidence intervals include 1.0 or cross zero, meaning non-significance.
Response 3: We thank the reviewer for this important observation. We acknowledge that, in some of the included studies, the confidence intervals for effect estimates include the null value, indicating statistical non-significance. In these cases, the expressions “associated with” or “no effect” were used as reported by the original study authors. To address this issue, we have now added a paragraph to the discussion highlighting the importance of critically interpreting the reported associations, especially in light of confidence intervals and statistical power. This addition can be found in lines 309–314.
Comments 4: Title clarity: “Association between maternal diet and frequency of micronuclei” → consider adding “in mothers and newborns” to make the scope clearer.
Response 4: We thank the reviewer for this helpful suggestion. We agree that specifying the population studied adds clarity to the title. Accordingly, we have revised the title to: “Association between maternal diet and frequency of micronuclei in mothers and newborns.”.
Comments 5: Grammar: Some sections use passive voice excessively. Consider active voice for clarity (e.g., “was observed” → “researchers observed”).
Response 5: We thank the reviewer for the valuable suggestion regarding the use of active voice to improve clarity. Our manuscript was carefully reviewed by professional language editors prior to submission to ensure linguistic accuracy and consistency.
Comments 6: Data Table 2: Please highlight significant findings visually (e.g., bold or asterisk).
Response 6: We thank the reviewer for this helpful suggestion. Significant findings have now been visually highlighted as recommended.
Comments 7: Conclusion section: The call for further research is clear, but recommendations for future study design (e.g., sample size calculation, use of biomarkers) should be more specific.
Response 7: We thank the reviewer for this insightful comment. In response, we have revised the Conclusion section to include more specific recommendations regarding future study design, such as detailed sample size calculations and the integration of objective biomarkers. These additions can be found in lines 385–403 of the revised manuscript.
Lines 385–403 now present clearer guidance for future research, emphasizing methodological rigor and the use of biomarkers to strengthen evidence.